# Molecular Cloning and Characterization of *MbMYB108*, a *Malus baccata* MYB Transcription Factor Gene, with Functions in Tolerance to Cold and Drought Stress in Transgenic *Arabidopsis thaliana*

**DOI:** 10.3390/ijms23094846

**Published:** 2022-04-27

**Authors:** Chunya Yao, Wenhui Li, Xiaoqi Liang, Chuankun Ren, Wanda Liu, Guohui Yang, Mengfei Zhao, Tianyu Yang, Xingguo Li, Deguo Han

**Affiliations:** 1Key Laboratory of Biology and Genetic Improvement of Horticultural Crops (Northeast Region), Ministry of Agriculture and Rural Affairs/National-Local Joint Engineering Research Center for Development and Utilization of Small Fruits in Cold Regions/College of Horticulture & Landscape Architecture, Northeast Agricultural University, Harbin 150030, China; 13753449701@163.com (C.Y.); lwh_neau@126.com (W.L.); a13989332297@163.com (X.L.); ren473475964@163.com (C.R.); yangguohui_neau@126.com (G.Y.); z1991689991@163.com (M.Z.); yty12980@163.com (T.Y.); 2Horticulture Branch of Heilongjiang Academy of Agricultural Sciences, Harbin 150040, China; haaslwd@126.com

**Keywords:** *Malus baccata* (L.) Borkh, *MbMYB108*, cold stress, drought stress

## Abstract

The MYB transcription factor (TF) family is one of the largest transcription families in plants, which is widely involved in the responses of plants to biotic and abiotic stresses, as well as plant growth, development, and metabolic regulation. In the present study, a new MYB TF gene, *MbMYB108*, from *Malus baccata* (L.) Borkh, was identified and characterized. The open reading frame (ORF) of *MbMYB108* was found to be 903 bp, encoding 300 amino acids. Sequence alignment results and predictions of the protein structure indicated that the MbMYB108 protein contained the conserved MYB domain. Subcellular localization showed that MbMYB108 was localized to the nucleus. The expression of *MbMYB108* was enriched in young and mature leaves, and was highly affected by cold and drought treatments in *M. baccata* seedlings. When *MbMYB108* was introduced into *Arabidopsis thaliana*, it greatly increased the cold and drought tolerances in the transgenic plant. Increased expression of *MbMYB108* in transgenic *A. thaliana* also resulted in higher activities of peroxidase (POD) and catalase (CAT), higher contents of proline and chlorophyll, while malondialdehyde (MDA) content and relative conductivity were lower, especially in response to cold and drought stresses. Therefore, these results suggest that *MbMYB108* probably plays an important role in the response to cold and drought stresses in *A. thaliana* by enhancing the scavenging capability for reactive oxygen species (ROS).

## 1. Introduction

MYB (v-myb avian myeloblastosis viral oncogene homolog) transcription factor (TF) is one of the most numerous and most versatile members of the plant TF family [1]. Its name is based on the fact that the encoded protein contains one or more highly conserved DNA domains—MYB domains [2]. The DNA-binding domains of MYB TFs contain an incompletely repeated R structure (R1, R2, and R3) composed of about 50 amino acid residues. MYB TFs can be divided into four subclasses according to the number of R structures contained in each gene [3]. Among them, the R1/R2-MYB subclass contains only one MYB domain, and the R2R3-MYB subclass contains two MYB domains, the R1R2R3-MYB subclass contains three continuous domains, and the 4R-MYB subclass contains four MYB domains [4,5,6].

A large number of previous studies have shown that MYB TFs regulate the expression of a series of genes by binding the cis-acting elements in the promoter region, and play an important role in the plant stress response [7]. After Paz-Ares cloned the *ZmMYBC1* gene related to pigment synthesis from maize [1], a large number of *MYB* genes were isolated and identified from plants, and it was discovered that *MYB* genes are widely involved in physiological and biochemical processes such as plant growth and secondary metabolism, and can also regulate many physiological responses of plants under biotic and abiotic stress. Expression of *OsMYB3R-2* in transgenic rice can enhance plant resistance to cold, dehydration, and salt stresses, while reducing sensitivity to abscisic acid (ABA) [8]. Overexpression of the rice R2R3-MYB TF gene *OsMYB55* enhances the metabolism of amino acids through transcriptional activation to enhance the resistance of rice to high temperatures [9]. In addition, the overexpression of *OsMYB4* will also significantly enhance the resistance of transgenic *Arabidopsis thaliana* to cold or freezing stresses [10]. Through the identification of 156 *GmMYB* genes in soybeans, it was found that the expression of 43 of them changed under the conditions of ABA, salt, drought, or cold stress [11]. Due to the above, MYB TFs participate in the response of plants to abiotic stress, and have potential application value in transgenic breeding for resistance to abiotic stresses, such as drought, cold, and low-temperature tolerances. According to previous research results, we speculate that there is also a MYB transcription factor gene in *Malus baccata* (L.) Borkh, which may change the tolerance of transgenic plants to abiotic stress by participating in the physiological and biochemical reactions.

*Malus baccata* is one of the commonly used grafting rootstocks of apple. It has strong grafting affinity between the same genus and a high survival rate. It is widely grown in the northeast and eastern inner Mongolia. Rootstocks are crucial to the morphogenesis of fruit trees and their adaptation to external conditions. Rational use of rootstocks is the most economical and effective measure to control tree crowns and achieve preterm and stable yields, which can further increase yield, improve quality, and improve land utilization [12]. Cold and drought resistances are important indicators for evaluating apple rootstocks [13]. However, most of the current apple rootstocks with high-quality characteristics are weak in cold and drought resistances [14]. From the transcriptome analysis of *M. baccata* seedlings under cold and/or drought stresses (results not presented here), we found that the *MbMYB108* expression level was significantly upregulated under both stresses. More importantly, through NCBI blast (BLAST: Basic Local Alignment Search Tool. Available online: https://blast.ncbi.nlm.nih.gov/Blast.cgi (accessed on 13 March 2021)) of the *MbMYB108* gene, we found that the closest *A. thaliana MYB* gene is *AtMYB96*, which is a famous MYB TF gene involved in drought stress through the ABA signaling pathway [15]. To better understand the role of MYB TF genes involved in cold and drought stresses, and to provide potential genetic resources for the improvement of the drought resistance of *Malus* plant, a new MYB TF gene was isolated from *M. baccata* and designated as *MbMYB108*. Moreover, it was found that the tolerance of transgenic *A. thaliana* to cold and drought stresses was increased because of the overexpression of *MbMYB108*.

## 2. Results

### 2.1. Isolation of MbMYB108 Gene from M. baccata

The ProtParam analysis (SIB Swiss Institute of Bioinformatics|Expasy. Available online: http://www.expasy.org/tools/protparam.html (accessed on 5 April 2021)) showed that *Mb**MYB108* cDNA was a complete open reading frame of 903 bp, encoding 300 amino acids (Appendix A). The predicted theoretical molecular mass of MbMYB108 was 33.939 kDa, with a theoretical isoelectric point of 7.82 and the average hydrophilicity coefficient was −0.753, which indicated that the protein is hydrophilic. Among the amino acids contained in MbMYB108 protein, Asn (10.4%), Ser (10.0%), Ala (8.0%), and Leu (7.0%) were relatively abundant.

### 2.2. Phylogenetic Relationship and Structural Prediction of MbMYB108 Protein

To explore the evolutionary relationship among plant MYB proteins, DNAMAN was used to compare MbMYB108 with 11 other MYB proteins of different species. Inside the red and blue frames was the conserved sequence of the amino acid of MYB proteins (Figure 1A), which was a characteristic sequence of the plant MYB TF family. The amino acid conserved sequence of MbMYB108 had high homology with the amino acid conserved sequences of the other 11 MYB proteins, but the non-conserved sequences had obvious differences, which was consistent with the characteristics of TFs, indicating that MbMYB108 belongs to the MYB TF family. The constructed phylogenetic tree showed that MbMYB108 and MdMYB108 (XP_008340570.2, *M. domestica*) had the highest homology (Figure 1B).

The predicted secondary structure of MbMYB108 contains predominant α-helices (37%) and coil structures (51%) (Figure 2A). The amino acid sequence of MbMYB108 had two SANT conserved domains (Figure 2B), presumably belonging to the R2R3-MYB TF family. The SWISS-MODEL online analysis suggests an overall α-helical structure for the MbMYB108 protein, consistent with the predicted secondary structure (Figure 2C).

### 2.3. MbMYB108 Was Localized in the Nucleus

In order to determine the specific location of MbMYB108 protein in cells, a fusion expression vector of green fluorescent protein (GFP) and the *MbMYB**108* gene was constructed. As shown in Figure 3, the *MbMYB108*-GFP fusion protein was targeted into the nucleus (Figure 3E), whereas the control GFP alone was distributed throughout the plasma membrane and the nucleus (Figure 3B). These results showed that MbMYB108 is a nucleus-localized protein.

### 2.4. Expression Level of MbMYB108 in M. baccata

The expression profile of *MbMYB108* in various *M. baccata* tissues under CK was investigated by using the qPCR assay. As shown in Figure 4A, *MbMYB108* was expressed at higher levels in young and mature leaves of *M. baccata* seedlings, but at lower levels in roots and stems. When subjected to low-temperature, high-salt, dehydration, high-temperature, and ABA treatments, the expression level of *MbMYB108* in young leaves of *M. baccata* increased quickly, reached a maximum at 3, 9, 7, 3, and 3 h, respectively, and then decreased (Figure 4B). The expression trend of *MbMYB108* in roots was consistent with that in young leaves, reaching a maximum at 7, 3, 5, 7, and 3 h, and then slightly decreasing (Figure 4C).

### 2.5. Overexpression of MbMYB108 Improves Cold Tolerance in Transgenic A. thaliana

To study the effect of overexpression of *MbMYB108* on cold and drought stresses in plants, *MbMYB108* was introduced into *A. thaliana*. Using WT and UL lines as controls, the T2 generation transgenic lines were analyzed by qPCR, and the results showed that the target gene had been integrated into the T2 generation transgenic lines (S1, S2, S3, S4, S5, S6) (Figure 5A). The three lines (S2, S4, S6) with a relatively high DNA expression level were selected to continue to cultivate to obtain T3 generation homozygous transgenic *A. thaliana*, which were used for subsequent research on morphological phenotype and physiological indexes.

As shown in Figure 5B, no significant difference in appearance was found among all *A. thaliana* lines (WT, UL, S2, S4, and S6) under CK (Cold 0 h). However, after dealing with cold (−6 °C) stress for 14 h (Cold 14 h), the transgenic lines (S2, S4, S6) looked much healthier than WT and UL. Under CK, there were no significant difference in the survival rate among all *A. thaliana* lines (WT, UL, S2, S4, and S6). However, when subjected to cold stress, the survival rates of WT and UL *A. thaliana* were only 20.5% and 15.8%, while the average survival rate of transgenic (S2, S4, S6) lines reached 84.9%. The survival rates of transgenic (S2, S4, S6) lines were significantly higher than those of WT and UL lines under cold treatment (Figure 5C).

To further understand the reasons why transgenic *A. thaliana* performed better under cold stress, the relevant physiological indexes of all lines (WT, UL, S2, S4, and S6) under normal temperature and cold stress were determined. The results showed that after cold treatment, chlorophyll content, proline content, POD, and CAT activities of transgenic (S2, S4, S6) lines were significantly higher than those of WT and UL, while MDA content and relative conductivity were significantly lower (Figure 5D–I). 

### 2.6. Overexpression of MbMYB108 Promotes the Expression of Cold Stress-Related Genes

The low-temperature signal transduction pathway dependent on CBF (CRT/DRE-binding factor) is a very important molecular regulation pathway in *A. thaliana* in response to cold stress. Therefore, we analyzed the expression level changes of several important genes, *AtCBF1*, *AtCBF3*, *AtRD29a,* and *AtCOR15a,* downstream of the MYB transcription factor under cold treatment (Figure 6). After 14 h of cold treatment at −6 °C, the expression level of the four genes was upregulated in all *A. thaliana* lines compared with CK, but the expression level of these four genes in *MbMYB108* overexpression transgenic lines (S2, S4, S6) was significantly higher than that in WT and UL lines, indicating that *MbMYB108* positively regulates *AtCBF1* and *AtCBF3*, thereby activating the expression of key genes *AtRD29a* and *AtCOR15a* under cold stress and improving the cold resistance of plants.

### 2.7. Overexpression of MbMYB108 Improves Drought Tolerance in Transgenic A. thaliana

In order to understand the tolerance of different *A**. thaliana* lines to drought stress, transgenic (S2, S4, S6), UL, and WT *A**. thaliana* with the same growth vigor were not watered for 10 days, and then the phenotype of each line was observed. As shown in Figure 7A, the transgenic (S2, S4, S6), WT, and UL lines all grew well under CK (drought 0 days). However, when watering was stopped for 10 days (drought 10 days), the transgenic (S2, S4, S6) *A. thaliana* had a better appearance than WT and UL lines. Similarly, under CK, there were no significant differences in the survival rates of all *A. thaliana* lines (WT, UL, S2, S4, and S6). However, after 10 days of drought stress, the survival rates of WT and UL lines were only 26.8% and 25.5%, while the average survival rate of transgenic (S2, S4, S6) lines was 75.0%. The survival rates of the transgenic (S2, S4, S6) *A. thaliana* under drought stress were significantly higher than those of WT and UL lines (Figure 7B).

Furthermore, the related physiological indexes of each line of *A**. thaliana* treated under CK and drought stress (stopped watering for 10 days) were determined. It was found that under drought stress, overexpression of *MbMYB108* resulted in lower MDA content and relative conductivity, higher chlorophyll and proline content, and higher POD and CAT activities in transgenic (S2, S4, S6) *A**. thaliana* relative to WT and UL lines. However, for the above indexes, there were no significant differences among entire test lines (WT, UL, S2, S4, and S6) under CK (Figure 7C–H).

### 2.8. Overexpression of MbMYB108 Promotes the Expression of Drought Stress-Related Genes

Drought stress can induce the increase of ABA levels in plants, and the ABA signal transduction activity is also enhanced. At the same time, studies have shown that ABA can also induce the expression of *MbMYB108*. Therefore, we selected the ABA synthesis gene *AtNCED3* and the ABA signal transduction-related gene *AtSnRK2.4*, and further analyzed their expression patterns in *MbMYB108* transgenic *A. thaliana* (Figure 8). The results showed that the expression levels of *AtNCED3* and *AtSnRK2.4* in *MbMYB108*-overexpressing lines (S2, S4, S6) were significantly higher than those in WT and UL lines under drought stress, indicating that *MbMYB108* positively regulates the expression of *AtNCED3* and *AtSnRK2.4*, and enhances drought tolerance in transgenic *A. thaliana* through ABA synthesis and signal transduction pathways. In addition, the expression levels of other drought stress-responsive genes *AtCAT1* and *AtP5CS* downstream of *MbMYB108* were also higher in transgenic (S2, S4, S6) lines under drought stress.

## 3. Discussion

There are many types of MYB family proteins with different functions. In 1987, Paz-Ares et al. cloned the *ZmMYBC1* gene by analyzing one genome and two cDNA clone sequences of maize [16], which is the first *MYB* gene to be identified in plants. In this experiment, *M**. baccata* was used as a test material, taking the nucleic acid sequence of *MdMYB108* (XM_008342348.3, *M**. domestica*) as a reference sequence, and primer 5.0 [17] software was used to design specific primers to amplify the target gene *MbMYB108*. The alignment of protein sequences found that MbMYB108 protein and other species of MYB proteins had high similarity in conserved sequences but large differences in non-conserved regions, which was an obvious feature of TFs. Analysis of the conserved domains revealed that MbMYB108 protein had two SANT conserved domain, which was characteristic of the MYB TF family [18], and it was speculated that MbMYB108 protein belongs to the R2R3-MYB family. Subcellular localization revealed that MbMYB108 is a nuclear-localized protein (Figure 3), which was consistent with other MYB proteins [19,20]. The phylogenetic tree indicated that MbMYB108 is most closely related to MdMYB108 (XP_008340570.2, *M**. domestica*) (Figure 1B).

In this study, the expression level of *MbMYB108* was the highest in the young leaves of *M. baccata*, while the expression level was less in the stems, which was only 1/9 of the expression level in the young leaves, indicating that its expression in the stems was inhibited or less relevant. In addition, the expression level of *MbMYB108* was unstable in different organs of *M. baccata*, suggesting that its expression pattern is tissue-specific. The present study showed that abiotic stresses such as low temperature, high salt, dehydration, high temperature, and ABA all induced the expression of *MbMYB108*, and the expression level of the same TF changed with time under different stress treatments. Among them, under cold and drought stresses, the upregulation of *MbMYB108* gene expression in young leaves and roots was the most obvious. Under cold stress, the expression levels of *MbMYB108* in young leaves and roots reached the peak at 3 and 7 h after treatment, and were 7.8 and 13.3 times higher than those in the untreated treatment, respectively. Under drought stress, the expression levels of *MbMYB108* in young leaves and roots peaked at 7 and 5 h after treatment, respectively, and were 7.2 and 12.9 times higher than those in untreated, respectively. This suggests that this gene may play an important role in the involvement of *M. baccata* in response to cold and drought stresses. It lays a foundation for further study on the functions of the *MbMYB108* gene.

After *MbMYB108* was introduced into the model plant *A. thaliana*, WT, UL, and transgenic (S2, S4, S6) *A. thaliana* were treated with cold and drought, respectively. After stress treatment, all lines showed a certain degree of damage, but the transgenic (S2, S4, S6) *A. thaliana* emerged with less yellowing and wilting, and statistical analysis showed that the survival rate of the transgenic (S2, S4, S6) *A. thaliana* was significantly higher than WT and UL lines. The results showed that overexpression of *MbMYB108* improves cold and drought resistances in transgenic (S2, S4, S6) *A. thaliana*. To further study the mechanism of this gene in the process of plant cold and drought resistances, the physiological and biochemical indexes and the expression of downstream stress-related genes of each *A. thaliana* line were determined and analyzed.

When plants are faced with cold stress, their cytoplasmic membrane permeability increases, electrolytes and soluble substances extravasate, and relative conductivity increases. Therefore, the level of relative conductivity can be used as an index for judging the cold resistance of plants [21]. Under cold stress, compared with WT and UL *A. thaliana*, the relative conductivity of overexpressed transgenic (S2, S4, S6) lines increased less, indicating that their cell membranes are less damaged. Chlorophyll is one of the important indexes for judging the cold resistance of plants. Low temperatures will decompose chlorophyll in plants, causing leaves to turn yellow, and therefore plants cannot perform photosynthesis normally, and even die [22]. Observing the phenotype of each line (WT, UL, S2, S4, and S6) under cold treatment, it was found that the degree of leaf yellowing of the transgenic line is lighter. Cold resistance can also be estimated by osmoregulation and antioxidant capacity [23]. Soluble sugars, proline, and soluble proteins are the main osmotic regulators in plant cytoplasm. They can increase the osmotic concentration of cells, lower the freezing point, buffer dehydration under cold stress, and help maintain the normal metabolism of cells [24]. Under cold stress, plants accumulate osmotic regulators such as proline [25], which enhances the water-holding capacity of cells, promotes protein hydration, and increases the content of soluble proteins [26], thereby maintaining enzyme activity in cold conditions. The proline content of the transgenic (S2, S4, S6) *A. thaliana* increased more under cold stress, indicating that they suffered less dehydration damage. Studies have shown that the activities of POD and CAT in the roots of cold-acclimated rice are significantly enhanced compared with ordinary rice [27]. Under cold stress, transgenic (S2, S4, S6) *A. thaliana* overexpressing *MbMYB108* had higher CAT and POD activities and lower MDA content compared with WT and UL. It indicated that the transgenic (S2, S4, S6) *A. thaliana* had a stronger ability to actively remove superoxide ions in the body under cold stress, and the tolerance to cold was also stronger.

When plants are subjected to chilling or freezing stress, the concentration of intracellular Ca^2+^ increases significantly, and the high concentration of intracellular Ca^2+^ is sensed by downstream signal receptors calmodulin (CAM) [28,29], calcium-dependent protein kinase (CDPK) [30], and calcium-interacting protein kinase (CIPKs) [31], and the signal is further transmitted. Studies have shown that MYB TFs can not only bind to the CBF promoter region to promote the expression of its downstream cold stress-related genes, but can also inhibit the expression of CBFs and negatively regulate the cold resistance of plants [32,33,34]. In addition, CBF can also induce the expression of CRT/DRE cis-acting elements of genes such as *CORs*, *RDs,* and *LTIs*, and improves plant cold resistance [35,36,37]. *MYB96* can bind to the promoter of *HHP* (transmembrane helix structure protein) gene to induce the synthesis of HHP protein, which in turn interacts with ICE1 (CBF gene expression inducer 1), ICE2, and CAMTA3 (calmodulin-binding transcriptional activator 3) to promote the expression of CBF, thereby activating the expression of downstream cold stress-related genes and improving the cold resistance of plants [38,39,40,41]. This present study quantitatively analyzed the expression levels of two key genes, *CBF1* and *CBF3,* in the CBF-dependent pathway, and their downstream cold stress responsive genes *COR15a* and *RD29a* under normal conditions and cold treatment, and found that *MbMYB108* can positively regulate *AtCBF1*, *AtCBF3*, *AtCOR15a*, and *AtRD29a* expression in response to cold stress through a CBF-dependent pathway.

Drought usually leads to excessive accumulation of reactive oxygen species (ROS), such as O_2_^2−^, H_2_O_2_, OH^−^, etc., in plants [42,43,44]. Most studies have shown that MYB TFs can participate in plant drought signal transduction and activate the ROS scavenging system, thereby avoiding the peroxidation of cell membrane lipids and the oxidative inactivation of metabolic enzymes, improving drought resistance of transgenic plants [45,46]. The birch *BplMYB46* gene binds to the MYBCORE and AC-box motifs and directly activates the expression of *SOD*, *POD,* and *P5CS* genes containing such elements, thereby reducing intracellular ROS levels, increasing proline content, and improving plant drought resistance [47]. The sweet potato *IbMYB116* gene was involved in the jasmonic acid (JA) signaling pathway to activate the ROS scavenging system, effectively reducing the oxidative damage of the plasma membrane caused by drought and other stress conditions, thereby improving the drought resistance of transgenic plants [48]. Compared with CK, transgenic (S2, S4, S6) *A. thaliana* exhibited higher CAT and POD activities, higher proline and chlorophyll content, and lower MDA content and relative conductivity than WT and UL lines under drought stress. The changes of these physiological indexes indicated that under drought stress, the transgenic (S2, S4, S6) lines could quickly scavenge reactive oxygen radicals in the body, reduce the damage to the plasma membrane, and maintain the integrity and stability of cells, and thus improve the drought resistance of plants.

Under drought stress, plants can induce stomatal closure by synthesizing ABA, thereby reducing water loss [49,50]. NCED (9-cis-epoxycarotenoid dioxygenase) plays an important role in ABA synthesis [51,52]. A study has shown that *AtNCED3* overexpression can increase ABA content in transgenic *A. thaliana* [53]. Under drought stress, the ABA content in transgenic kidney beans increased due to the expression of *NCED1*, and drought resistance of the overexpressed line was significantly improved [54]. The ABA signal transduction pathway is also an important pathway for plants to respond to drought stress [55,56]. Under drought stress, related TFs bind to downstream target genes to regulate their expression and generate intercellular signal ABA. ABA receptors sense the accumulation of ABA and bind to it, thereby inhibiting the activity of intracellular PP2C (protein phosphatase 2C) [57,58], resulting in an increase in SnRK (sucrose non-catalytic protein kinase 2) content [59], and downstream TFs are phosphorylated, regulating ABA-related gene expression, response to drought stress, and then stomatal closure [60,61]. In this study, the expression levels of *AtNCED3* and *AtSnRK2.4* were more significantly increased in transgenic (S2, S4, S6) *A. thaliana* after drought stress compared with WT and UL lines, indicating that *MbMYB108* could positively regulate the expression of *AtNCED3* and *AtSnRK2.4*. It was further shown that *MbMYB108* can not only regulate ABA synthesis, but also participates in the regulation of ABA signal transduction genes, and jointly regulates the response of transgenic (S2, S4, S6) *A. thaliana* to drought stress through these two pathways. In addition, we also analyzed the expression of drought stress-responsive genes *AtCAT1* and *AtP5CS*, and found that the expression levels of both genes were significantly increased in transgenic (S2, S4, S6) *A. thaliana* under drought stress. These results suggest that *MbMYB108* can promote the expression of downstream drought stress-related genes through multiple pathways and improve the drought tolerance of plants.

## 4. Materials and Methods

### 4.1. Plant Materials and Growth Conditions

Rapid propagation of *M. baccata* tissue culture seedlings was performed in MS growth medium (MS + 0.6 mg/L of 6-benzylaminopurine (6-BA) + 0.6 mg/L of indolebutyric acid (IBA)) for 30 days. Then, robust tissue culture seedlings were selected and transferred to MS rooting medium (MS + 1.2 mg/L IBA) to continue culturing until white roots grew [62]. Finally, the tissue culture seedlings with new roots were then transferred to Hoagland nutrient solution [63] for cultivation, and the new hydroponic solution was replaced as needed during hydroponics. The temperature of the culture chamber was kept at around 25 °C, and the relative humidity was kept at 80–85%. When the tissue culture seedlings grew 7–9 true leaves and the root system was well-developed, 60 seedlings with good shape and basically the same growth were selected for sampling. First, unexpanded young leaves, completely unfolded mature leaves, phloem at the second and third node stem segments, and newly emerged roots were sampled in 10 of the *M.*
*baccata* seedlings, respectively. The remaining 50 seedlings were divided into 5 groups for different stress treatments: low-temperature treatment (hydroponic seedlings were cultured at 4 °C), salt treatment (hydroponic seedlings were cultured under high-salt conditions of 200 mM NaCl), dehydration treatment (hydroponic seedlings were cultured in Hoagland nutrient solution with a concentration of 20% PEG6000), high-temperature treatment (hydroponic seedlings were cultured at 38 °C), and ABA treatment (hydroponic seedlings were cultured in hydroponic solution with an ABA concentration of 50 μM), and the seedlings were cultured under normal Hoagland nutrient solution as control, young leaves, and roots of control, and all treatments were sampled. The samples of all control and treated seedlings were sealed after treatments of, respectively, 0, 1, 3, 5, 7, 9, and 12 h, immediately frozen in liquid nitrogen, and stored at −80 °C for RNA extraction.

### 4.2. Isolation and Cloning of MbMYB108

Extraction of *M. baccata* total RNA was performed with the OminiPlant RNA Kit (Kangweishiji, Beijing, China). The first-strand cDNA was synthesized using TransScript^®^ First-Strand cDNA Synthesis SuperMix (TransGen Biotech, Beijing, China). A pair of primers (*MbMYB-108F* and *MbMYB-108R,* Appendix A) were designed based on the homologous regions of *MdMYB108* (XM_008342348.3, *Malus domestica*) to amplify the full-length cDNA sequence. The whole sequence of *MbMYB108* was obtained by polymerase chain reaction (PCR) with primers, and the first-strand cDNA of *M. baccata* was used as a template. The target fragments were separated by agarose gel electrophoresis [64]. The obtained DNA fragment of the *MbMYB108* gene was gel-purified and cloned into the pEASY^®^-T1 vector (TransGen Biotech, Beijing, China) [65] and sequenced (Beijing Genomics Institute, Beijing, China).

### 4.3. Sequence Analysis and Structure Prediction of MbMYB108

Multiple sequence alignment of MbMYB108 and MYB TFs from other species was performed with DNAMAN 5.2. The phylogenetic tree was constructed by the neighbor-joining method [66] with MEGA 7 (Home. Available online: http://www.megasoftware.net (accessed on 5 April 2021)) [67]. ExPASy (ExPASy-ProtParam tool. Available online: https://web.expasy.org/protparam/ (accessed on 20 April 2021)) was used to predict the primary structure of MbMYB108 protein. The domain of MbMYB108 protein was predicted on the SMART website (SMART: Main page. Available online: http://smart.embl-heidelberg.de/ (accessed on 20 April 2021)), using SWISS-MODEL (SWISS-MODEL. Available online: https://swissmodel.expasy.org/ (accessed on 20 April 2021)) to predict the tertiary structure of MbMYB108 protein.

### 4.4. Subcellular Localization Analysis of MbMYB108 Protein

The *MbMYB108* ORF was cloned into the *BamH*I and *Sal*I sites of the pSAT6-GFP-N1 vector using primers (*site*-F and *site*-R, Appendix A) with *BamH*I and *Sal*I restriction sites to construct the *MbMYB108*-GFP transient expression vector. The recombinant plasmid containing the *MbMYB108* gene was injected into onion epidermal cells by particle bombardment [68]. After overnight culture, the fluorescence signals of the control protein GFP and the *MbMYB108*-GFP fusion protein were observed under confocal microscopy (LSM 510 Meta, Zeiss, Germany). The nucleus was marked using DAPI staining.

### 4.5. Quantitative Real-Time PCR (qPCR) Expression Analysis of MbMYB108

DNAMAN was used to perform sequence alignment to find out the conserved regions of the *MbMYB108* nucleic acid sequence and select the sequences with high specificity to design qPCR [69,70] primers (*MbMYB*-*108*qF and *MbMYB-108*qR, Appendix A), and *Actin* gene (NC_024251.1, *M. domestica*) primers (*Actin*-F and *Actin*-R, Appendix A) were sent for synthesis. Using cDNA of pretreated *M. baccata* material as a template, the expression level of the *MbMYB108* gene was detected using *TransStart*^®^ Green qPCR SuperMix (TransGen Biotech, Beijing, China) according to the manufacturer’s protocol. PCR conditions were as follows: 94 °C for 30 s, 40 cycles of 95 °C for 5 s, 54 °C for 40 s, 72 °C for 30 s, and then 72 °C for 10 min. Analysis of relative transcript level data was carried out using the 2^−ΔΔCT^ method [71].

### 4.6. Generation of Transgenic A. thaliana Overexpressing MbMYB108

Primers (*HF* and *HR*, Appendix A), including target fragment-specific primer sequences, restriction sequences (*BamH*I and *Sal*I), and overlapping sequences, were designed to amplify target fragments. Then, according to the principle of homologous recombination (ClonExpress^®^II One Step Cloning Kit, Vazyme, Nanjing, China), the target fragment was ligated into the PCAMBIA2300 vector to construct the PCAMBIA2300-*MbMYB108* overexpression vector. The *MbMYB108* gene was introduced into columbia-0 ecotype *A. thaliana* by *Agrobacterium*-mediated transformation of GV3101 using the inflorescence infusion method [72,73,74,75]. Transgenic *A. thaliana* was selected on MS medium containing 50 mg/L of Kanamycin until generation T3, and the T3 generation plants were used for further analysis.

### 4.7. Stress Treatment and Determination of Related Physiological Indexes in A. thaliana

Wildtype (WT), empty vector line (UL, line transformed with empty vector only), and T3 transgenic lines (S2, S4, S6) of *A. thaliana* were sown in 1/2 MS medium, respectively, and after 10 days, seedlings that had cotyledons and had grown well were transferred to nutrient pots (4 plants per pot, the composition of substrate was turfy soil:vermiculite = 2:1). Each line of *A. thaliana* was divided into two groups: one group was treated with cold stress (−6 °C for 14 h) and the other group was treated with drought stress (stopped watering for 10 days), and then they were returned to normal culture for 7 days to remove stress. Finally, morphological changes were observed, and survival rates were calculated [76].

The materials of each line of *A. thaliana* under CK and after stress treatment were collected, and their physiological indexes were determined, respectively. Chlorophyll content was measured using the immersion extraction method [77]. According to the method of Huang et al., proline was extracted by the sulfosalicylic acid method and its content was determined [78,79,80]. Relative conductivity was measured using the pump-down method [81]. The guaiacol method was used to determine peroxidase (POD) activity [82,83], the ultraviolet (UV) absorption method was used to measure catalase (CAT) activity [84], and the spectrophotometer color method was used to measure malondialdehyde (MDA) activity [85,86].

### 4.8. Expression Analysis of MbMYB108 Downstream Genes

Using *AtActin* as the internal reference, the mRNAs of each line of *A. thaliana* under CK and after stress were extracted and reverse transcribed into first-strand cDNA. The qPCR experiments were performed on several important regulatory genes downstream of MYB TFs: cold stress response key genes (*AtCBF1, AtCBF3, AtCOR15a, AtRD29a*) and drought stress response key genes (*AtNCED3, AtSnRK2.4., AtCAT1, AtP5CS*), using specific primers (Appendix A).

### 4.9. Statistical Analysis

SPSS software was used to analyze the differences with Duncan’s multiple range tests. Statistical differences were referred to as significant when * *p* ≤ 0.05 and ** *p* ≤ 0.01.

## 5. Conclusions

In the present study, a new *MYB* gene was isolated from *M. baccata* and named *MbMYB108*. Subcellular localization showed that MbMYB108 protein was located in the nucleus. When *MbMYB108* was introduced into *A. thaliana*, it increased the levels of proline and chlorophyll, and improved the activities of POD and CAT, but decreased MDA content and relative conductivity, especially under cold and drought treatments. Taken together, our results suggest that *MbMYB108* plays an important role in the response to cold and drought stress by enhancing the capability of scavenging ROS.

## Figures and Tables

**Figure 1 ijms-23-04846-f001:**
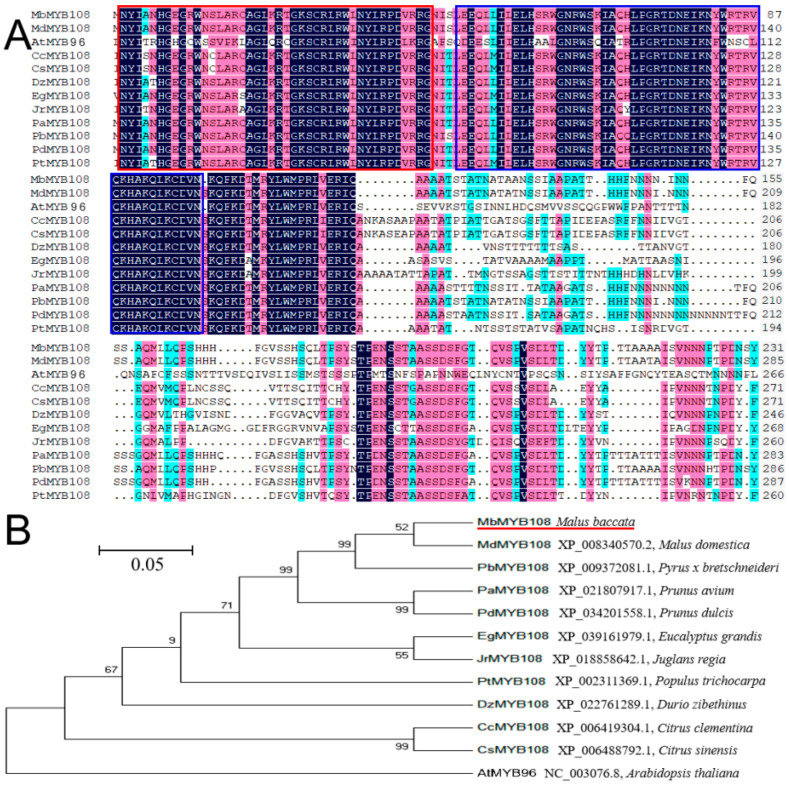
Comparison and phylogenetic relationship of MbMYB108 with other species of MYB transcription factor (TF) proteins. (**A**) Amino acid sequence alignment of MbMYB108 with other species of MYB TF proteins. Conserved domains are shown in red and blue boxes. (**B**) Phylogenetic tree analysis of MbMYB108 and other species of MYB TF proteins.

**Figure 2 ijms-23-04846-f002:**
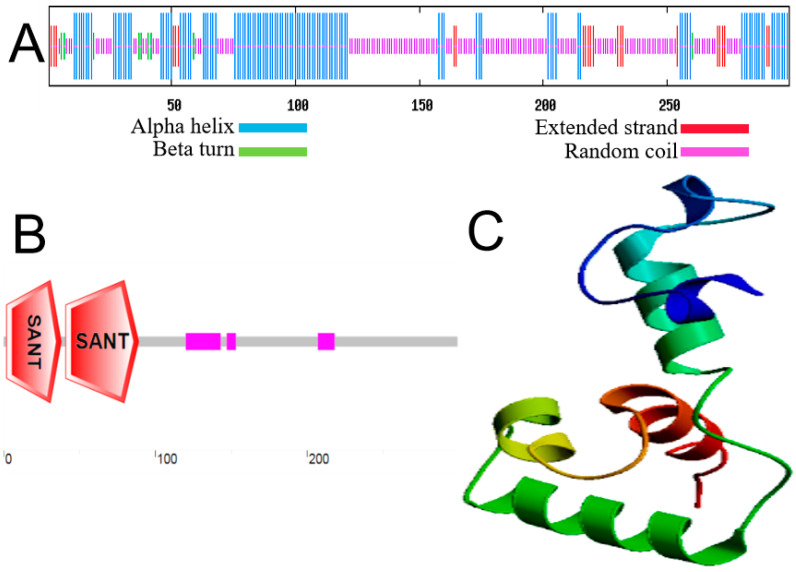
Structural prediction of MbMYB108 protein. Prediction of the secondary structure (**A**), domains (**B**), and tertiary structure (**C**).

**Figure 3 ijms-23-04846-f003:**
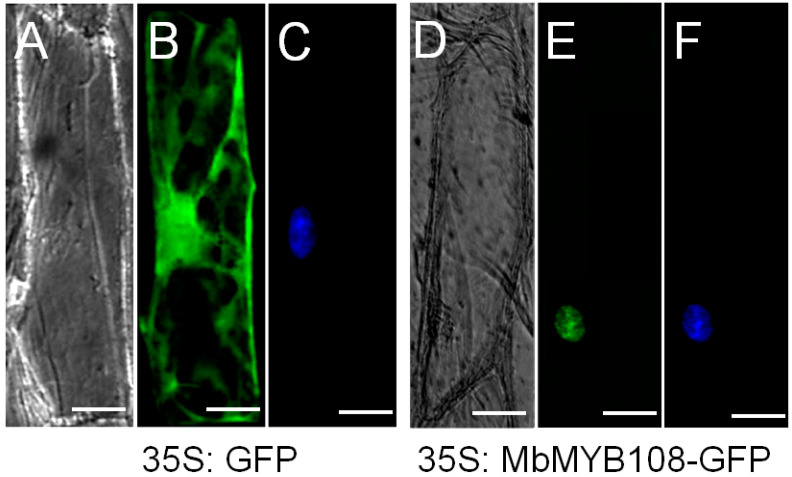
Subcellular localization of MbMYB108 protein. Transient expressions of green fluorescent protein (GFP) and *MbMYB108*-GFP fusion protein in onion epidermal cells was observed by fluorescence microscopy: (**A**,**D**) were taken under bright light, (**B**,**E**) were taken under dark field, and (**C**,**F**) are the results of DAPI staining for 24 h. Scale bar corresponds to 5 μm.

**Figure 4 ijms-23-04846-f004:**
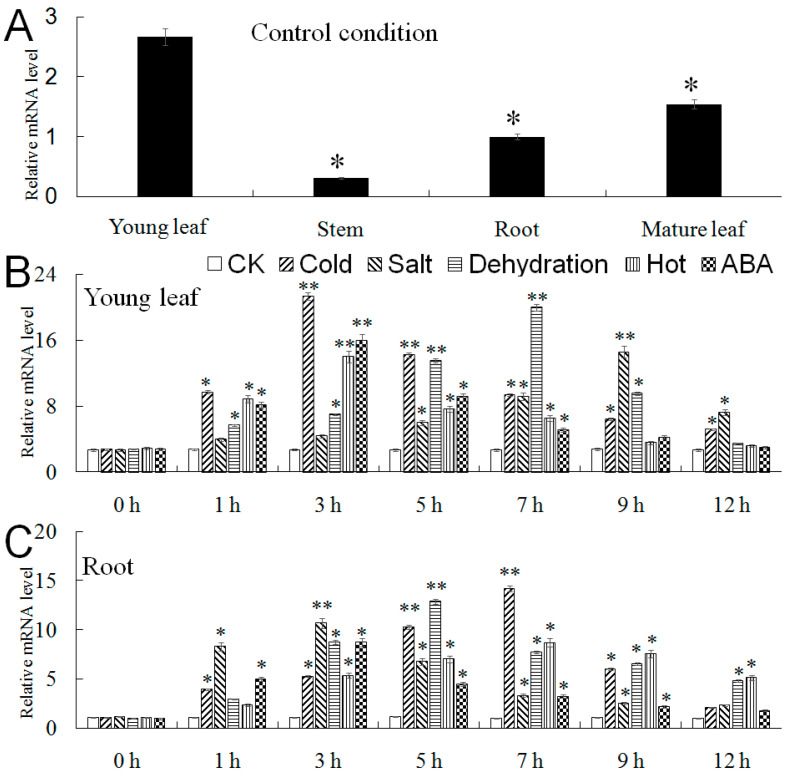
Time-course expression patterns of *MbMYB**108* in *Malus baccata*. (**A**) Expression patterns of *MbMYB**108* in young leaves (partly expanded), mature leaves (fully expanded), roots, and stems in normal condition (room temperature and normal watering). The expression level of young leaves was used as a control. Expression patterns of *MbMYB**108* in control condition (CK), low-temperature (4 °C), high-salt (200 mM NaCl), dehydration (20% PEG), high-temperature (38 °C), and ABA (50 μM) stress in young leaves (**B**) and roots (**C**) at the following time points: 0, 1, 3, 5, 7, 9, and 12 h. Data represent the mean of three replicates. Error bars represent standard deviation. Asterisks above error bars indicate significant differences between treatment and control (0 h) (* *p* ≤ 0.05, ** *p* ≤ 0.01).

**Figure 5 ijms-23-04846-f005:**
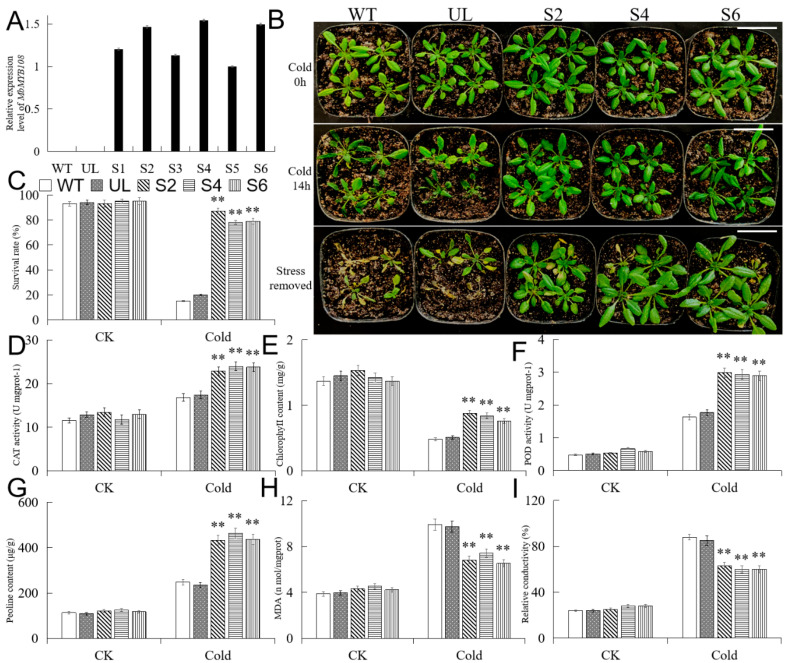
Overexpression of *MbMYB108* in *Arabidopsis*
*thaliana* improved cold tolerance. (**A**) Transgenic *A. thaliana* qPCR validation. (**B**) Phenotypic map of each line (WT, UL, S2, S4, and S6) of *A. thaliana* under CK (Cold 0 h), cold stress (Cold 14 h), and recovery conditions (stress removed). Scale bar corresponds to 4 cm. (**C**) The survival rates of each line (WT, UL, S2, S4, and S6) of *A. thaliana* under CK and cold stress. (**D**) Catalase (CAT) activity. (**E**) Chlorophyll content. (**F**) Peroxidase (POD) activity. (**G**) Proline content. (**H**) Malondialdehyde (MDA) content. (**I**) Relative conductivity. Asterisks above the error bars indicate extremely significant differences between transgenic (S2, S4, S6) and WT *A. thaliana* (** *p* ≤ 0.01). The level of each index in the WT line was used as a control.

**Figure 6 ijms-23-04846-f006:**
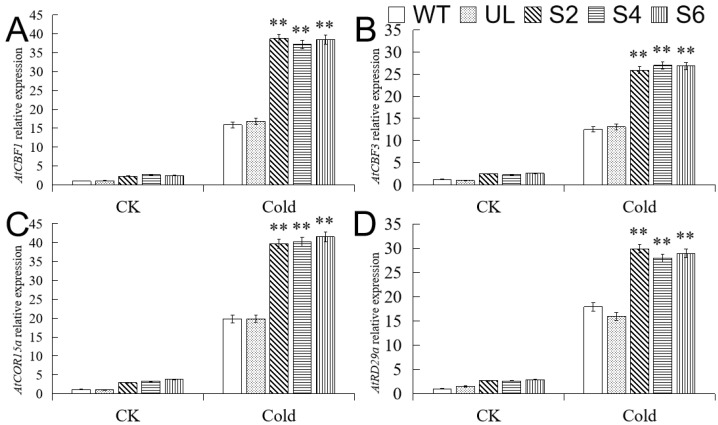
Relative expression levels of cold stress-related genes in WT, UL, and transgenic *A. thaliana.* Relative expression levels of *AtCBF1* (**A**), *AtCBF3* (**B**), *AtCOR15a* (**C**), and *AtRD29a* (**D**). Data represent the mean of three replicates. Error bars represent standard deviation. Asterisks above error bars indicate significant difference compared to the WT line (** *p* ≤ 0.01).

**Figure 7 ijms-23-04846-f007:**
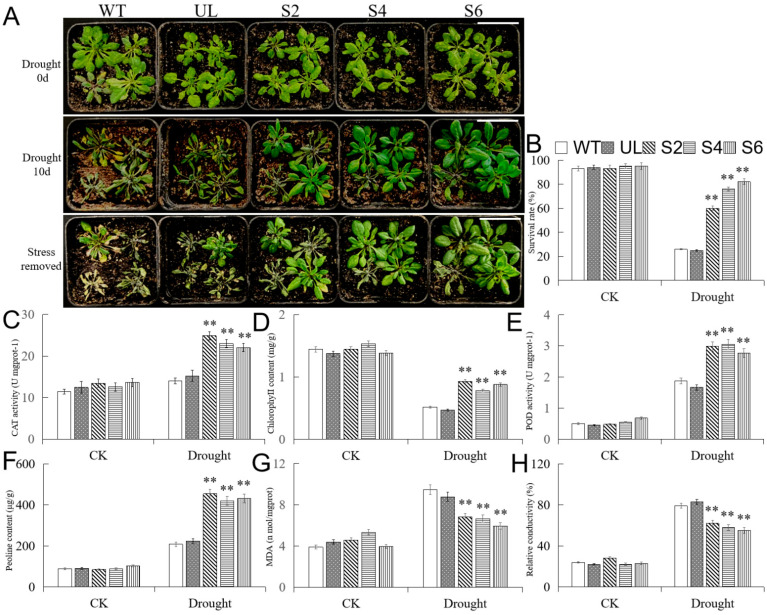
Overexpression of *MbMYB108* in *A. thaliana* improved drought tolerance. (**A**) Phenotypic map of each line (WT, UL, S2, S4, and S6) of *A. thaliana* under CK (Drought 0 days), drought stress (Drought 10 days), and recovery conditions (stress removed). Scale bar corresponds to 4 cm. (**B**) The survival rates of each line (WT, UL, S2, S4, and S6) of *A. thaliana* under CK and drought stress. (**C**) Catalase (CAT) activity. (**D**) Chlorophyll content. (**E**) Peroxidase (POD) activity. (**F**) Proline content. (**G**) Malondialdehyde (MDA) content. (**H**) Relative conductivity. Asterisks above the error bars indicate extremely significant differences between transgenic (S2, S4, S6) and WT *A. thaliana* (** *p* ≤ 0.01). The level of each index in the WT line was used as a control.

**Figure 8 ijms-23-04846-f008:**
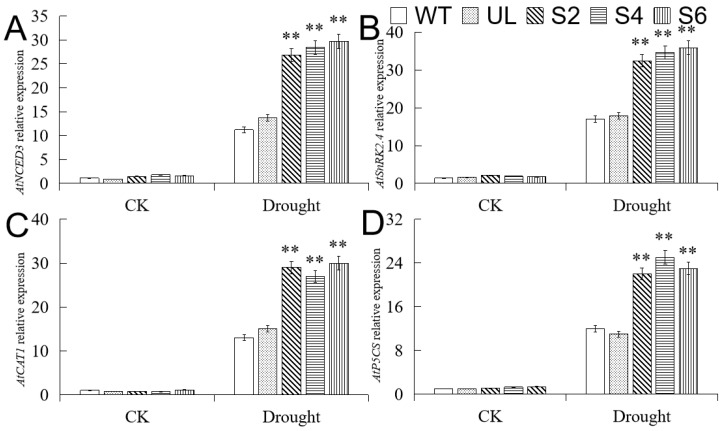
Relative expression levels of drought stress-related genes in WT, UL, and transgenic *A. thaliana.* Relative expression levels of *AtNCED3* (**A**), *AtSnRK2.4* (**B**), *AtCAT1* (**C**), and *AtP5CS* (**D**). Data represent the mean of three replicates. Error bars represent standard deviation. Asterisks above error bars indicate significant difference compared to the WT line (** *p* ≤ 0.01).

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
