# Peer review of "Molecular Cloning and Characterization of MbMYB108, a Malus baccata MYB Transcription Factor Gene, with Functions in Tolerance to Cold and Drought Stress in Transgenic Arabidopsis thaliana"

_ijms, 2022, doi:10.3390/ijms23094846_

Round 1
Reviewer 1 Report
This is a well conducted and rather well analyzed work on the enhanced abiotic stress resistance of Malus baccata by the MbMYB108 transcription factor protein. Even the direct measurement of ROS scavenging is lacking, the enzymatic measurements sustain the involvement of MbMYB108 in the abiotic stress protection of the plant. Results dealing with changes in the soluble protein profiles should be interesting with respect to a possible evolution of some drought and cold resistance markers in plants overexpressing the MbMYB108 transcription factor. These points should be discussed in the paper.
Minor points:
Lines 113-114 :
Replace by “The predicted secondary structure of MbMYB108 contains predominant a-helices (37%) and coil structures (51%) (Figure 2A).”
Lines 116-17 :
Replace by “The SWIS-MODEL online analysis suggests an overall a-helical structure for the MbMYB108 protein, consistent with the predicted secondary structure (Figure 2C).”
Line 174:
Replace by “… no significant phenotypic difference was ….”
Author Response
Reviewer 1
This is a well conducted and rather well analyzed work on the enhanced abiotic stress resistance of Malus baccata by the MbMYB108 transcription factor protein. Even the direct measurement of ROS scavenging is lacking, the enzymatic measurements sustain the involvement of MbMYB108 in the abiotic stress protection of the plant. Results dealing with changes in the soluble protein profiles should be interesting with respect to a possible evolution of some drought and cold resistance markers in plants overexpressing the MbMYB108 transcription factor. These points should be discussed in the paper.
Response: We appreciate your valuable suggestions and advices on our manuscript. I think they are very helpful and important, and revisions had been made in the manuscript accordingly. Modified parts are marked in red in the manuscript.
We have supplemented the part about the changing of soluble protein in plants under drought and cold stresses in discussion. Your suggestion for dealing with the results of changes in the soluble protein profile is very useful, we ignored this content due to the problem of experimental design, we will add research on this content in future experiments.
Here I would like to response the comments and add some explanations as follows.
Minor points:
- Lines 113-114: Replace by “The predicted secondary structure of MbMYB108 contains predominant a-helices (37%) and coil structures (51%) (Figure 2A).”
Response: Yes, we have accepted your suggestion and modified this part.
- Lines 116-117: Replace by “The SWISS-MODEL online analysis suggests an overall a-helical structure for the MbMYB108 protein, consistent with the predicted secondary structure (Figure 2C).”
Response: Yes, we have accepted your suggestion and modified this part.
- Line 174: Replace by “… no significant phenotypic difference was ….”
Response: Yes, we have rechecked and found the sentence redundant and deleted it.
Reviewer 2 Report
In the manuscript titled: Molecular Cloning and Characterization of MbMYB108, a Malus baccata MYB Transcription Factor Gene, with Functions in Tolerance to Cold and Drought Stress in Transgenic Arabidopsis thaliana. Authors suggest that MbMYB108 can regulate key genes to increase the ability to remove reactive oxygen species (ROS) under stress, thereby enhancing transgenic plants' cold and drought resistance. The manuscript needs professional English editing and stresses the points mentioned below.
Abstract
Abstract is carelessly written authors should incorporate their notable findings and adequately connect with the sentences they choose to correspond.
Introduction
- The introduction section must have a clear hypothesis and significantly develop the second paragraph of your manuscript. Make it more connecting to the problem statement.
- Overall there is the repetition of the information, which could be avoided.
Discussion
- This section should include more information and references related to the relevant and related works.
Conclusions
- If possible, restructure and carefully edit the conclusion section and add clear information regarding the most noteworthy findings.
Author Response
Reviewer 2
In the manuscript titled: Molecular Cloning and Characterization of MbMYB108, a Malus baccata MYB Transcription Factor Gene, with Functions in Tolerance to Cold and Drought Stress in Transgenic Arabidopsis thaliana. Authors suggest that MbMYB108 can regulate key genes to increase the ability to remove reactive oxygen species (ROS) under stress, thereby enhancing transgenic plants' cold and drought resistance. The manuscript needs professional English editing and stresses the points mentioned below.
Response: We appreciate your valuable suggestions and advices on our manuscript. I think they are very helpful and important, and revisions had been made in the manuscript accordingly. Modified parts are marked in red in the manuscript.
Here I would like to response the comments and add some explanations as follows.
The language of manuscript have also been revised.
Major Concerns:
- Abstract
Abstract is carelessly written authors should incorporate their notable findings and adequately connect with the sentences they choose to correspond.
Response: Yes, we have accepted your suggestion and modified this part.
- Introduction
The introduction section must have a clear hypothesis and significantly develop the second paragraph of your manuscript. Make it more connecting to the problem statement. Overall there is the repetition of the information, which could be avoided.
Response: Yes, we have accepted your suggestion and modified this part.
- Discussion
This section should include more information and references related to the relevant and related works.
Response: Yes, we have accepted your suggestion and modified this part.
- Conclusions
If possible, restructure and carefully edit the conclusion section and add clear information regarding the most noteworthy findings.
Response: Yes, we have accepted your suggestion and modified this part.
Round 2
Reviewer 2 Report
Manuscript is now suitable for publication.